# Effects of Multi-Task Mode on Cognition and Lower Limb Function in Frail Older Adults: A Systematic Search and Review

**DOI:** 10.3390/healthcare11233012

**Published:** 2023-11-21

**Authors:** Cenyi Wang, Bingqing Wang, Jiling Liang, Ziru Niu, Aming Lu

**Affiliations:** 1School of Physical Education and Sports Science, Soochow University, Suzhou 215006, China; cywang11@suda.edu.cn (C.W.); wangbingqing202210@163.com (B.W.); qdcynzr@163.com (Z.N.); 2Graduate School, Wuhan Sports University, Wuhan 430079, China; ljl19930210@163.com

**Keywords:** frail older adults, multi-task, dual-task, cognition, lower limb function

## Abstract

The application of multi-tasking (MT), especially dual-tasking (DT), in frail older adults is currently gaining attention. The aim was to review the application of the MT mode on cognition and lower limb function in frail older adults, including the MT test and MT training. By searching five electronic databases, Scopus, PubMed, PEDro, Web of Science and the Chinese electronic database, a total of 18 studies were finally included in this study, with 7 articles on MT testing and 11 articles on MT training. The results of the study showed that the current testing and training of MT is mainly based on the DT mode, with a wide variety of test types and protocols, as well as a variety of outcomes. The included studies suggested that DT can be used as a test to assess cognitive and lower limb function in the frail population and that an MT (DT) training program with an intervention period of ≥3 months or a duration of ≥60 min per session could improve cognitive and lower limb function in the frail population, thereby reducing the risk of falls. Further research is required to explore the effects of different types of MT and task prioritization in frail older adults.

## 1. Introduction

Frailty is a complex clinically relevant syndrome characterized by the deterioration of multiple physiological systems associated with reduced organ function due to sarcopenia, malnutrition and altered hormone levels [1,2]. Frailty has not yet been directly defined as a disease. In clinical studies, the Fried criteria, Kihon checklist, frailty index and other measurement tools are widely used to assess frailty [3,4,5]. However, the weakened stress-regulating capacity of the body caused by frailty, such as reduced activity, fatigue and physical performance, greatly increases the risk of adverse events in the elderly population [6]. Studies have shown that the global population of older adults over the age of 65 suffering from frailty, sarcopenia and cognitive impairment is projected to increase from 700 million in 2019 to 1.5 billion in 2050 [7]. As human life expectancy increases, aging and age-related conditions such as frailty and muscle atrophy have a significant impact on the function and quality of daily life in the elderly population [8,9]. How to effectively detect or prevent the aging process of the organism through various interventions and enhance the physical activity capacity of the elderly population has become the focus of scientific attention.

In daily life, people are commonly faced with multiple tasks at the same time, such as talking while walking and walking while holding a glass of water, mobile phone, or food. Conventional rehabilitation training or assessment measures are mostly physical function rehabilitation and testing under single task conditions, which only aim to improve or check the movement ability of elderly patients in a real-time environment while ignoring the movement or cognitive dysfunction that may exist in elderly patients under multi-task (MT) conditions [10]. Simply returning elderly patients to community life after a period of single-task testing or training may not effectively improve the quality of daily life in the elderly population but rather increase the incidence of elderly patients who fall in MT situations due to overconfidence in their ability to perform physical activity. Tombu et al. [11] suggested that the dual-task (DT) mode may distract the participants’ attention from the first task and that the behavioral ability in this environment might be more representative. Raffegeau et al. [12] showed that various types of DT modes can seriously deteriorate the gait performance of Parkinson’s disease (PD) patients. MT training can improve the postural control of older adults in daily activities, thus enabling them to cope better with complex living environments, which is vital for preventing falls and improving the quality of life of older individuals in the future. A systematic review of PD found that adding DT to gait and balance training can further benefit the gait and balance function in patients with mild to moderate PD compared with a single task [13]. Another study also showed that DT training can improve gait speed, stride length and cognitive function in patients with neurological impairment, thereby contributing to increased patient independence [14]. At the current time, the application of the DT mode in PD, Alzheimer’s disease and other neurological disorders has been widely reported, and in recent years, some scholars in the fields of psychology and geriatrics have started to concentrate on the larger population of frail older people.

Previous studies have demonstrated that a significant age-related decline in muscle mass and strength in frail older adults produces a large number of patients with sarcopenia [15]. Similar to frailty, sarcopenia is strongly associated with decreased physical activity, fractures and even falls in older adults [16]. As the global population ages, frailty and sarcopenia are becoming more prevalent in elderly individuals. Unless detected early and treated effectively, this condition will have a significant adverse impact on the health and quality of life of the elderly population. Studies have shown that skeletal muscle activity has powerful immune and redox effects that can alter brain function and reduce muscle metabolism, and that aging may play an essential role in the deterioration of skeletal muscle function and cognitive decline [17]. Simultaneously, gait performance and cognition share common neural pathways and DT training at different intensities can improve performance, such as cognitive function and gait performance [18,19]. Studies have shown that DT exercise training can significantly improve physical weakness and cognitive function, reduce the risk of falls and even reverse the development of sarcopenia in the elderly population [20]. However, current evidence on the effects of the MT mode on skeletal muscle strength and function is limited, and the relationship between cognition and skeletal muscle function has not been fully elucidated [21].

Therefore, this study aims to summarize the application of different types of MT modes as exercise interventions and assessment measures in frail older adults to provide more evidence for optimized exercise interventions to improve the physical mobility of frail older adults, prevent the occurrence of falls and reduce the incidence of disability and mortality.

## 2. Materials and Methods

In this paper, a systematic literature search and review of research on dual-task activities in frail older adults was conducted [22].

Five electronic databases, including four English databases (Scopus, PubMed, PEDro and Web of Science) and one Chinese electronic database (China National Knowledge Infrastructure), were searched from their inception to November 2023. The selected search terms included “frail elderly”, “frail older adults”, “sarcopenia”, “dual-task”, “multi-task”, “limb function” and “motor function”, “muscle function”, “balance”, “postural control”, “balance control”, “physical performance”, “cognition”, and “cognitive function”. A variety of search strategies was used systematically, and reviewers made the final literature inclusion by reading abstracts and full-text content. The literature search included all intervention studies, and any training and testing studies on MT were included in the initial search. Details of our search strategy are available in Appendix A.

Only full articles in English or Chinese were available for further evaluation. Indicators of interest in this study include the results of all experiments involving lower limb function and cognitive function, and studies that do not focus on frail older adults are excluded from this review. The reviewer extracted information from all final included literature, including author, age of subjects, sample size, intervention type, MT time, indicator results, etc. All information is summarized, synthesized and presented in table form.

## 3. Results

### 3.1. Study Inclusion

A total of 2728 articles were identified through an extensive literature search. After removing duplicates, 839 studies were retained. Further screening was conducted by reviewing the title and abstract of the literature, and 185 articles were obtained. After reading the full text of 185 articles, a total of 18 articles were selected following different routes of the process, of which 7 were related to MT testing and 11 were related to MT training (see Figure 1).

### 3.2. Conceptual Description

All studies [23,24,25,26,27,28,29] reporting MT testing in the frail elderly population used the DT mode; with seven studies using the cognitive-motor DT mode, including a walking task while performing a counting task, walking while talking or naming an animal, etc.; and one study [29] additionally selecting the moto-motor DT mode (walking while holding a glass of water). All included studies reported results for lower limb functional indicators, such as gait speed, turn duration, step length and step frequency; and two of the studies [24,29] presented gait indicator results by reporting dual-task costs (DTCs). The results of four studies [23,25,27,28] showed that DT testing significantly worsens gait performance in frail older adults; and both the studies by Piche et al. and Zheng et al. showed that the DTC of gait was significantly higher in the frail population [24,29]. Cognitive function outcomes were reported in two studies; and the indicators chosen were call accuracy and counting accuracy. Moreover, the results of one study [28] showed that the DT test significantly reduced counting accuracy in a group of frail older adults. However, Ansai et al. revealed that the DT test did not show a significant difference in potentially frail older adults with cognitive impairment compared to the control group (Table 1).

A total of 11 studies [20,30,31,32,33,34,35,36,37,38,39] were finally included. All studies chose the cognitive-motor MT mode, and most of the intervention programs were in the form of DT balance training combined with cognitive tasks such as counting, memorizing, naming and video games, etc. Only one study [38] reported the use of MT training (e.g., walking to music and responding to changes in the rhythmic patterns of the music). In terms of intervention duration, six studies [20,30,33,34,35,39] reported intervention durations of 3 to 6 months, four studies [31,32,36,37] reported intervention durations shorter than 3 months, and one MT training study [38] reported a 6-month intervention. Meanwhile, in terms of intervention frequency, 10 study protocols used a frequency of one to three times per week, and only 1 study [39] opted for DT training five times a week. However, it is noteworthy that only three articles [33,35,39] described the intensity of training (low and moderate intensity), which was not reported in the remaining studies. Of the studies that included MT training, all reported on muscle function and balance, with handgrip strength and gait speed being the main measures of muscle function and the time up and go test (TUGT) and the short physical performance battery test (SPPB) being the main measures of balance. Among the studies reporting results on muscle function and balance, most demonstrated that MT training significantly improved balance and muscle function in frail older adults, but five studies [30,31,35,36,37] found no significant differences in some of the muscle function indicators (handgrip strength, gait speed and 6-min walking test), and three studies [30,36,37] found no significant differences in some of the balance indicators (TUGT, SPPB, inclination angle and Tinetti test). In addition, nine studies [20,30,31,33,34,35,36,37,39] reported on cognitive function, mainly using the Montreal Cognitive Assessment (MoCA) and Mini-Mental State Examination (MMSE) as indicators, and only a few studies [34,35,36] used the verbal reaction time, rate of response and verbal fluency test to assess cognitive outcomes (Table 2).

### 3.3. Characteristics of the Scope and Population

The seven DT test studies contained a total of 489 subjects, 105 males and 289 females (one study did not report the gender of the subjects [23]), with female subjects dominating the studies. Six studies involved participants over the age of 70, two of whom were over the age of 80 [27,28], and one involved participants under the age of 70 [29]. Regarding the publication year of the studies, all six studies were published within the decade, and four of them [23,24,28,29] were published in the past 5 years, which suggests that the use of DT testing in frail populations has gradually been attracting attention in recent years (Table 1).

Eleven MT training studies included a total of 911 subjects, six studies [30,33,34,36,37,39] did not report the gender of the subjects, and the remaining studies included a total of 95 male and 320 female subjects. With regard to the age of the subjects, four studies [30,32,35,37] had subjects aged 80 years or older, three studies [31,38,39] involved subjects over 70 years, and the remaining four studies only mentioned subjects over the age of 60 years. Concerning the year of the study publication, among the included studies, eight studies [20,30,31,32,33,34,35,39] related to MT training were published in the past 5 years, and the application of MT training in frail older adults has received more attention from researchers in recent years (Table 2).

## 4. Discussion

### 4.1. Efficacy of MT Testing in Assessing Balance, Muscle and Cognitive Function in Frail Older Adults

MT and especially DT are now widely recognized and used in neuropsychology and have been used to study the degree of independence between different independent task processes or the process of sharing neural resources between different tasks where there are interactions. The DT paradigm requires the body to perform multiple tasks simultaneously, which requires the optimization of “neural resources” compared to a single task; thus, the DT paradigm is considered an evolutionary advantage in the development of the human nervous system, which is reflected in various types of activities in our daily lives [40].

Maintaining the postural stability of the human body during various physical activities is a complex process that requires the integration of inputs from multiple sensory systems, such as vision, vestibular and proprioception; these sensory inputs are combined with appropriate neuromuscular responses and the flexibility of joint movements to achieve postural balance [41]. Studies have shown that the extent to which a DT task affects postural stabilization depends on factors such as age and task type [42]. Of the dual-task testing studies included in this paper, all subjects were over 70 years of age, with the exception of the study by Zheng et al. [29]. All studies used cognitive-motor DT as a test protocol to assess motor function in a population of frail older adults, with only one study using an additional walking-with-a-glass-of-water DT to assess gait cadence. Considering that motor DTs, such as walking with a glass of water, do not demand much of the subjects’ functional brain resources and require fewer cognitive resources compared to cognitive tasks, DTs such as counting and naming may be more challenging for the elderly population, especially those with physical or cognitive dysfunctions [43]. Safe walking is an attention-demanding task for older adults, requiring high levels of mobility and cognitive flexibility to adapt to a range of environmental demands, such as controlling the walking direction and visual target recognition and tracking [44]. Walking was chosen as the motor task in the DT in almost all of the included DT study protocols, and the results of the study indicated that, compared with the older adult population without frailty or physical and cognitive decline, the frail older adult population completed the DT with a reduced gait speed, an increase in the number of steps, a support phase, a double-support phase, a total walking time and a turning duration, as well as a significantly higher percentage of lateral line stepping over and stops. Functional activity performance in DT walking may more accurately reflect the ability of the frail individual to perform multiple tasks simultaneously and could be more valuable for identifying postural control and the risk of falling in the elderly population.

However, it is noteworthy that one study [26] found no significant differences in gait performance and call accuracy between older adults at risk of falling with mild cognitive impairment and older adults with preserved cognition in completing the TUGT and calling a telephone number for the DT. In clinical studies, commonly used outcome measures for DT gait analysis are readily available parameters such as gait speed and step frequency. Nevertheless, simple timing devices and manually recognized information may not provide a complete picture of the true gait performance of a debilitated population. Clinical assessment of patients using DTC parameters, which can quantify DT and thereby reduce physiological differences in gait between individuals, may be advantageous [45]. Two of the current included studies [24,29] used DTC parameters to analyze various commonly used gait metrics and showed that the DTC was significantly higher in frail older adults than in the non-frail population. Laurence et al. [46] suggested that the mechanisms of aging not only affect sensory and motor systems but also that executive functions, cortical information processing, attentional resource allocation and attentional capacity are all altered in older adults as the body ages. Therefore, the exploration of physical functioning in frail older adults requires consideration of the influence of multiple factors from motor, cognitive and other sources.

### 4.2. Effects of MT Training on Balance, Muscle and Cognitive Function in Frail Older Adults

Falls are severe adverse events that affect the safety and quality of life of the elderly population, and while muscle function and proprioception decline in the debilitated elderly population, the impact of cognitive function-related mechanisms such as executive function and attention on falls cannot be ignored. The older female population comprised a very large proportion of subjects in the current inclusion of studies related to MT testing and training. Considering that the loss of skeletal muscle mass and bone mass may be more pronounced in women than in older men due to the postmenopausal decline in estrogen, older women are at a significantly higher risk of developing sarcopenia and osteoporosis in old age, and the risk of falls in older women is greatly increased compared to that in men [47,48].

At present, a large amount of evidence has confirmed that cognitive-motor DT exercise training can significantly improve physical activity function in people with cognitive dysfunction, such as those with Alzheimer’s disease [49,50]. According to Varela-Vasquez et al. [51], simultaneous DT exercise training is an effective way to improve motor performance and body coordination. In this review, all included studies used cognitive-motor MT for intervention training in a population of frail older adults. The specifics of the intervention training methods varied, but the types of exercise training chosen were mostly focused on balance, aerobic and resistance exercises. Although only three studies [33,35,39] mentioned the intensity of the intervention training, the descriptions of the study protocols from all the included literature showed that the exercise intensity was mostly low to moderate. Meanwhile, in terms of intervention duration, most of the studies reported an intervention duration of more than 3 months, and the interventions were effective on cognitive, muscle and balance function in the frail older population. However, three studies [31,36,37] with intervention cycles of less than 3 months and less than 60 min per session showed that MT training did not significantly improve handgrip strength and gait speed in frail older adults compared to conventional exercise interventions. Considering shorter cycles and training durations, fewer than three weekly doses of cognitive-motor MT exercises may not be sufficient to improve human motor performance in multi--tasking. An important principle of motor learning is the frequent and repetitive training of specific motor tasks to continuously improve motor performance; hence, repetitive practice of sufficient frequency and duration is required to fully develop cerebral neuroplasticity [52].

Additionally, Trombetti et al. [38] investigated the effects of music-based MT training on muscle and balance function in an elderly population at high risk of falling, and their results showed that MT training for more than 6 months significantly reduced gait variability and improved postural balance in older adults performing single-task and DT tasks. It has been suggested that this factor of improved gait variability in the elderly population through a period of MT or DT practice could be related to the development of more automated tasks and task coordination skills [36,53]. Additionally, the effects of MT on cognitive function still require adequate assessment through more detailed work, such as neuropsychological batteries. Interestingly, Mak et al. [32] chose analogy training (i.e., “imagine you are kicking a ball”) for their study of an elderly population at high risk of falling and found that analogy training appeared to have a more favorable effect on walking performance than single-task and DT effects. However, the DT training protocol for this study only selected walking while performing a computational task, whereas the DT protocol test selected a Stroop task with a clock rest, and it is worth considering whether the choice of index test method may have somewhat influenced the results of the study (see Figure 2).

The fixed and variable priority of DT training has also been the focus of many researchers. Because the transferability performance of DT training may be limited, it is clearly crucial for MT training to select and evaluate whether the training task is more in line with the intended skill or motor performance of the tested subject. Silsupadol et al. [36] reported the effects of 4 weeks of fixed-priority and variable-priority DT training on a community-based group of older adults with balance impairment. The results of the study indicated that a variable priority training strategy was more effective in improving balance and cognitive performance in an older population than singletasking and a fixed priority. Silsupadol and colleagues also noted that the positive effects of variable priority training could be a result of the development of walking task automation and task integration skills [54].

Regarding the limitations of this review, the heterogeneity of the current included studies was large, with different types of tasks, task difficulty and testing methods chosen, and the gender of the samples was mostly focused on older women. All results and conclusions remain to be analyzed with more careful consideration.

## 5. Conclusions

DT testing can assess cognitive and lower limb function in frail older adults, and an MT (DT) training program with a specific intervention cycle (≥3 months) or intervention duration (≥60 min) could improve cognitive and lower limb function in the frail population, thereby reducing the risk of falls. Considering the diversity of daily activities, further research is recommended to investigate the effects of different types of MT and task prioritization in the frail population to improve the benefits of training.

## Figures and Tables

**Figure 1 healthcare-11-03012-f001:**
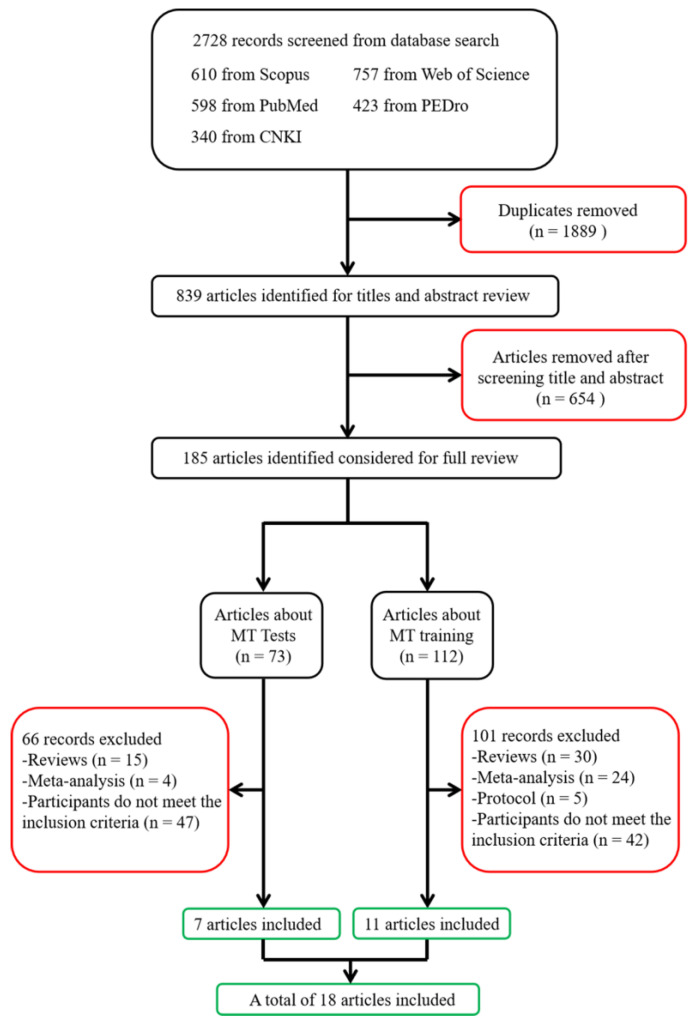
Flow diagram of the study selection. CNKI, Chinese National Knowledge Information Database.

**Figure 2 healthcare-11-03012-f002:**
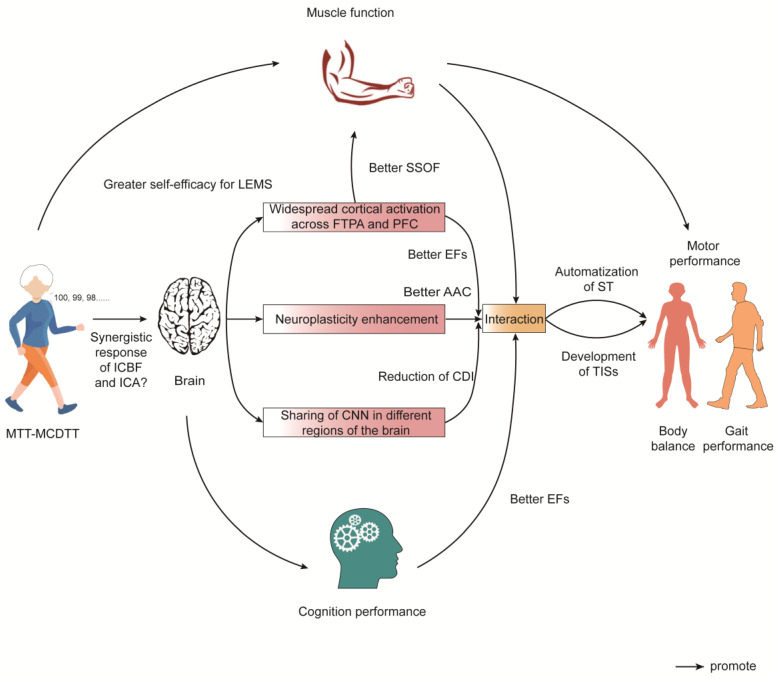
Potential mechanisms of the effects of MT training on frail older adults (AAC, attentional allocation capacity; CDI, competitive demands interference; CNN, complex neural networks; EFs, executive functions; FTPA, fronto-temporo-parietal areas; ICA, increase in cerebral angiogenesis; ICBF, increase in cerebral blood flow; LEMS, Lambert–Eaton myasthenia syndrome; MCDTT, motor-cognitive dual task training; MTT, multitask training; PFC, prefrontal cortex; SSOF, sensory system organs function; ST, single task; TISs, task-integration skills).

**Table 1 healthcare-11-03012-t001:** Characteristics of the included test studies.

Author (Year, Country)	Target Population	Sample (Gender Ratio)	Age (Year)	Test Content	Outcomes
Beauchet et al. [27] (2005, Switzerland)	Transitionally frail older adults	30 (M: 3; FE: 27)	82.60 ± 7.10	D1: Walking + Counting backward D2: Walking + Naming animals	G1 ↓
Guedes et al. [25] (2014, Brazil)	Older people with or without frailty	27 (PF): 27 (F): 27 (NF) (M: 18; FE: 63)	PF: 70.11 ± 7.30 F: 75.48 ± 7.08 NF: 69.60 ± 5.45	Walking + Talking	Gait speed ↓
Ansai et al. [26] (2017, Brazil)	Older people with or without cognitive problems (with falling experience and gait speed <1.0 m/s)	40 (MCI): 40 (PC) (M: 22; FE: 58)	MCI: 75.80 ± 6.30 PC: 73.50 ± 6.20	TUGT + Calling phone number	Call accuracy - TUGT-DT -
Toosizadeh et al. [28] (2019, America)	Older adults with or without mild cognitive impairment (partially pre-frail or frail)	34 (MCI): 35 (NMCI) (M 29; FE: 40)	MCI: 83.88 ± 6.57 NMCI: 83.83 ± 6.92	UEFM + Counting backward (−1 or −3)	Motor function speed ↓ Accuracy ↓
Piche et al. [24] (2022, France)	Older people with or without frailty	31 (PF): 16 (F): 19 (NF) (M: 24; FE: 42)	75.50 ± 6.50	Walking + Counting (−3)	DTC (G2) ↑
Zheng et al. [29] (2022, China)	Older adults with or without mild cognitive impairment (TUGT >1 0 s and MoCA < 24)	38 (MCI): 30 (NMCI) (M: 9; FE: 59)	MCI: 60.90 ± 7.30 NMCI: 60.10 ± 6.70	D1: Walking + Holding a bottle of water D2: 8UGT + Computation	DTC (gait cadence) ↑
Zhou et al. [23] (2023, China)	Frail older people with or without sarcopenia	44 (FS): 51 (F) (Not applicable)	≥75	D1: Walking + Counting backwards D2: Walking + Counting (−3) D3: Walking + Naming animals	Gait speed ↓ Turn duration ↑

Note: TUGT, time up and go test; MoCA, Montreal cognitive assessment; M, male; FE, female; PF, pre-frail; F: frail; NF, non-frail; MCI, mild cognitive impairment; PC, preserved cognition; NMCI, non-mild cognitive impairment; FS, frail with sarcopenia; D1, the first dual-task group; D2, the second dual-task group; UEFM, upper-extremity function measurement; 8UGT, the 8-foot up and go test; D3, the third dual-task group; G1, walking time, number of steps, percentage of lateral line stepping-over and stops; TUGT-DT, timed up and go test associated with a motor-cognitive task; DTC, dual-task cost; G2, gait speed, step length, step length variability, stance and swing phase time, single and double support, cadence, step time variability and gait speed variability; ↑, raise and improve; ↓, decrease; -, no statistically significant effects.

**Table 2 healthcare-11-03012-t002:** Characteristics of the included training studies.

Author (Year, Country)	Target Population	Sample Size (Gender)	Age (Year)	Dual-/Multi-Task Exercise				Outcomes
				Content	Duration	Frequency	Intensity	
Silsupadol et al. [36] (2009, America)	Community dwelling older adults with balance impairment	EG (DT-FP): 8 EG (DT-VP): 6 CG (ST): 7 (Not appliable)	≥65	Balance training + Cognitive tasks (counting backward, naming objects, and spelling words backward)	4 weeks	3 times per week 45 min per time	Not mentioned	Muscle function: Gait speed: DT-VP vs. ST - DT-FP vs. ST - Cognition: Verbal reaction time: DT-VP vs. ST ↓ DT-FP vs. ST↓ Rate of response: DT-VP vs. ST↑ DT-FP vs. ST↑ Balance capacity: Inclination angle: DT-VP vs. ST↓ DT-FP vs. ST -
Trombetti et al. [38] (2011, Switzerland)	Community dwelling older adults at increased risk of falling	EG (MMT): 66 (M: 2, F: 64) CG (DI): 68 (M: 3, F: 65)	MMT: 75 ± 8 DI: 76 ± 6	Various multitask exercises (e.g., walking in time to the music and responding to changes in the music’s rhythmic patterns)	6 months	1 time per week 60 min per time	Not mentioned	Muscle function: Gait speed: (EG, CG)↑ Balance capacity: TUGT: (EG, CG)↓ Tinetti test: (EG, CG)↑ EG vs. CG↑
Szturm et al. [37] (2011, Canada)	Frail community dwelling and ambulatory older adults	EG (DT): 14 CG (ST): 13 (Not applicable)	EG: 80.5 ± 6 CG: 81 ± 7	Dynamic balance exercises coupled with video game play	8 weeks	2 times per week 45 min per time	Not mentioned	Muscle function: Gait speed: (EG, CG) - Cognition: MMSE: (EG, CG) - Balance capacity: TUGT: (EG, CG) - EG vs. CG ↓ BBS: (EG, CG) ↑ EG vs. CG ↑
Rezola-Pardo et al. [35] (2019, Spain)	Physically and/or cognitively impaired older adults	DT: 42 (M: 13, F: 29) MC: 43 (M: 15, F: 28)	84.90 ± 6.70	Multicomponent Exercise (strength and balance exercises) + Cognitive tasks (counting/remembering words)	3 months	2 times per week 60 min per time	moderate intensity	Muscle function: SFT: (DT, MC) ↑ 6-MWT: DT, -; MC, ↑ Cognition: MoCA: (DT, MC) - VF: (DT, MC) - Balance capacity: TUGT: DT, -; MC, ↓ SPPB: (DT, MC) ↑
Bischof et al. [30] (2021, Germany)	Long-term nursing home residents	EG (DT): 12 CG (ST): 12 (Not applicable)	83.70 ± 6.40	Balance training + Naming/Counting backward	16 weeks	2 times per week 45–60 min per time	Not mentioned	Muscle function: HGS: (EG, CG) - Gait speed: EG, ↑; CG, - Cognition: MoCA: (EG, CG) - Balance capacity: SPPB: (EG, CG) -
Merchant et al. [20] (2021, Singapore)	Possible sarcopenia	EG (DT): 111 (M: 30, F: 81) CG (ST): 40 (M: 21, F: 19)	≥60	Personalized DT incorporating resistance, balance, aerobic+ Cognitive tasks (e.g., marching, clapping, with step-up/down movement on the step-board with simultaneous naming/recalling tasks)	3 months	1 or 2 times per week 60 min per time	Not mentioned	Muscle function: HGS: (EG, CG) ↑ Gait speed: EG, ↑; CG, - Cognition: MoCA: EG, ↑; CG, - cMMSE: EG, ↑; CG, -Balance capacity: SPPB: EG, ↑; CG, -
Merchant et al. [33] (2021, Singapore)	Older adults at risk (pre-frail, frail or sarcopenia)	DT: 197 (Not applicable)	≥60	Aerobics, resistance, balance exercises + Counting (±3)/Remembering (the numbered ladder)	3 months	1 or 2 times per week 60 min per time	Low-to-moderate intensity	Muscle function: Gait speed: ↑ Cognition: MoCA: ↑ cMMSE: ↑ Balance capacity: SPPB: ↑
Kwan et al. [31] (2021, China)	Frail elderly at an elderly community center	EG (DT): 9 (M: 1, F: 8) CG (ST): 8 (M: 1, F: 7)	EG: 73.00 ± 7.50 CG: 77.50 ± 15.30	VR (training in daily living tasks) partially simultaneous motor-cognitive training (finding someplace, grocery shopping, birdwatching, etc.)	8 weeks	2 times per week 30 min per time	Not mentioned	Muscle function: HGS: (EG, CG) - Cognition: MoCA: EG, ↑; CG, - Balance capacity: TUGT: EG, -; CG, ↓
Zhao et al. [39] (2021, China)	Possible sarcopenia	EG (DT): 20 CG (ST): 21 (Not applicable)	71.49 ± 7.43	Treadmill training + Counting (±3)/Naming	12 weeks	5 times per week 25 min per time	At the initial speed of 1.98 km/h	Muscle function: ASMI: (EG, CG) ↑ HGS: (EG, CG) ↑ 6-MWT: (EG, CG) ↑ Cognition: MoCA: (EG, CG) ↑ EG vs.VG ↑ Balance capacity: TUGT: (EG, CG) ↑ POMA: (EG, CG) ↑ EG vs.VG ↑ m-FES: (EG, CG) ↑ EG vs.VG ↑
Mak et al. [32] (2022, China)	Community dwelling older adults at risk of falling (POMA < 24)	DT: 27 (M: 8, F: 19) AG: 25 (M: 4, F: 21) ST: 19 (M: 12, F: 7)	DT: 81.96 ± 6.51 AG: 82.38 ± 6.36 ST: 80.11 ± 7.08	Walking + Counting	4 weeks	3 times per week 45 min per time	Not mentioned	Muscle function: Gait speed: ST ↑; DT -; AG ↑ Balance capacity: TUGT: (ST, DT, AG) ↑ POMA: (ST, DT, AG) ↑ BBS: (ST, DT, AG) * (AG appeared to be superior to DT and ST)
Nascimento et al. [34] (2023, Brazil)	Older females at risk of falling	EG (DT): 22 CG (ST): 22 (Not applicable)	EG: 66.14 ± 4.15 CG: 66.27 ± 4.04	Walking/One-leg standing/+ Counting/Memorization/Visual tasks and word spelling	12 weeks	2 times per week 60 min per time	Not mentioned	Muscle function: LEMS: (EG, CG) - EG vs.VG ↑ Cognition: VF: (EG, CG) ↑ EG vs.VG ↑ Balance capacity: TUGT: (EG, CG) - EG vs.VG ↓

Note: POMA, performance oriented mobility assessment; EG, experimental group; CG, control group; DT, dual task; MC, multicomponent group; FP, fixed-priority; VP, variable-priority; ST, single task; MMT, music-based multitask training; DI, delayed intervention; M, male; F, female; AG, analogy; TUGT, time up and go test; MMSE, mini-mental state examination; HGS, hand grip strength; ASMI, average skeletal muscle mass index; 6-MWT, 6-min walking test; SFT, senior fitness test; MMoCA, Montreal cognitive assessment; SPPB, short physical performance battery test; cMMSE, Chinese mini-mental state examination; m-FES, modified fall efficacy scale; BBS, Berg balance scale; LEMS, lower limb muscle strength; VF, verbal fluency test; ↑, indicates a significant increase; ↓, indicates a significant decrease; -, indicates no significant difference.

## Data Availability

Data are contained within the article and Appendix A.

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
