# Peer review of "Effects of Multi-Task Mode on Cognition and Lower Limb Function in Frail Older Adults: A Systematic Search and Review"

_healthcare, 2023, doi:10.3390/healthcare11233012_

Round 1
Reviewer 1 Report
Comments and Suggestions for Authors
Comments to the Author
1. General comments
The study summarizes the effects of multi-task training on frail older adults. It's an interesting topic, but there are major concerns about the method of review. Please consider the following.
2. Specific comments
i) What type of review is this? I don't believe it's a systematic review, but is it a scoping or narrative review? Either way, make sure you follow the guidelines for your review. There was no description of compliance with reporting guidelines in the text. Less reproducible reviews are of poor value.
ii) Line 95-99 lists search terms. Please include all official search formulae as an appendix. Also, please include the official date of the search. I will search and see if I can get similar results if necessary.
iii) How many people were screened? Consider following the guidelines and adding the s details of the process.
iv) The study finally incorporates 18 articles. Isn't meta-analysis possible? It is recommended to implement.
Author Response
General comments
The study summarizes the effects of multi-task training on frail older adults. It's an interesting topic, but there are major concerns about the method of review. Please consider the following.
- Specific comments
- i) What type of review is this? I don't believe it's a systematic review, but is it a scoping or narrative review? Either way, make sure you follow the guidelines for your review. There was no description of compliance with reporting guidelines in the text. Less reproducible reviews are of poor value.
Response: Thank you very much for your patient review! This review is a systematic literature search and review of research on dual-task activities in the frail elderly and is not a systematic review. At present, there are many categories of reviews. Some scholars divide them into the following categories according to review strategies, research quality assessment, and analysis methods: narrative review, descriptive review, scoping review, qualitative systematic review, meta-analytic review, umbrella review, narrative review, descriptive review, scoping review, qualitative systematic review, meta-analytic review, umbrella review, realist review, theoretical review, critical review. Realist review, theoretical review, critical review (Pare, Trudel, Jaana, & Kitsiou, 2015). Some researchers divide the review into 14 types according to the content and method of research, including: critical review, literature review, mapping review, meta-analytic review, rapid review, state-of-the-art reveiw, systematic review, systematic search and review, umbrella review, etc (Grant, & Booth, 2009). This manuscript belongs to a systematic search and review. Meanwhile, we recognize that the adoption of the PRISMA guidelines provides assistance in drafting a high-quality review report, and if you feel this is necessary, we will immediately follow up by adding the appropriate form as an attachment.
- ii) Line 95-99 lists search terms. Please include all official search formulae as an appendix. Also, please include the official date of the search. I will search and see if I can get similar results if necessary.
Response: Thank you for your advice. I have included the official search formula as an appendix (Appendix 1).
iii) How many people were screened? Consider following the guidelines and adding the s details of the process.
Response: Two authors (CW, BW) were involved in this systematic search. Due to the length and type of the study, we have not included details of the process in the text of the manuscript. The search process is as follows: the retrieved studies were screened simultaneously and independently by two reviewers (CW, BW) based on previously established inclusion criteria. The extraction content mainly includes the first author, publication date, target population, age, gender, sample size, test/intervention protocol, etc. After the unqualified and duplicate studies were eliminated, the remaining studies were read and rescreened in full. If the information provided is unclear or disputed, the corresponding author of the study will be contacted for a thorough investigation. Throughout the process, all disagreements between reviewers were resolved through discussion. If you feel that this is necessary to include in the manuscript, we will add it immediately in the follow-up. Thank you again for your advice!
- iv) The study finally incorporates 18 articles. Isn't meta-analysis possible? It is recommended to implement.
Response: Thank you for your advice. There are many related indicators in the literature, including scales and some biomechanical measurement indicators. If some indicators are proposed for further meta-analysis, the number of included literatures for most indicators may be less than three, which will make the meta-analysis lack evidence-based significance. Thus, we only classified the indicators of the included literatures, analyzed the different types of indicators and results, and summarized the potential mechanisms (Figure 2).
Reviewer 2 Report
Comments and Suggestions for Authors
Minor questions
- Line 60 and 389: (De Freitas Tb Ms et al., 2020): please, review the text citation and references. The authors' training acronyms were inserted in the text and references (MS; Master of Science; P.T. Physiotherapist; PhD, Philosophy Doctor. Review and delete, please)
- Please, review the lack of space between the word and the parentheses, there are many cases throughout the text (Raffegeau et al.(Raffegeau et; adults(Lee; performance(Grande; population(Merchant; elucidated(Peng et al., 2020); lives(Piqueres-Juan et al., 2021; ...
- Line 295: what is AD patients? Please, explain in full in the text
- References:
- Line 379: Neurophysiologie clinique = Clinical neurophysiology 48, 361-375
- Line 396: Fadzil, S.N.M., Osman, I., Ismail, S., Hashim, M.J.M., Khamis, M.R., 2021. Does financial support improve the well-being of the 396 elderly?, PROCEEDINGS OF 8TH INTERNATIONAL CONFERENCE ON ADVANCED MATERIALS ENGINEERING & 397 TECHNOLOGY (ICAMET 2020)
- Line 402, 410, ... : several article titles with initial letters in capital letters while others in lower case (please standardize)
Author Response
- Line 60 and 389: (De Freitas Tb Ms et al., 2020): please, review the text citation and references. The authors' training acronyms were inserted in the text and references (MS; Master of Science; P.T. Physiotherapist; PhD, Philosophy Doctor. Review and delete, please)
Response: Thank you for your careful review! We apologize due to a formatting error when inserting references in Endnote. We have made changes (highlighted in yellow).
- Please, review the lack of space between the word and the parentheses, there are many cases throughout the text (Raffegeau et al.(Raffegeau et; adults(Lee; performance(Grande; population(Merchant; elucidated(Peng et al., 2020); lives(Piqueres-Juan et al., 2021; ...
Response: Thank you very much for your patient review! We checked it carefully and made changes (highlighted in yellow).
- Line 295: what is AD patients? Please, explain in full in the text
Response: We have made changes (highlighted in yellow).
- References:
- Line 379: Neurophysiologie clinique = Clinical neurophysiology 48, 361-375
Response: We apologize due to a formatting error when inserting references in Endnote. We have made changes.
- Line 396: Fadzil, S.N.M., Osman, I., Ismail, S., Hashim, M.J.M., Khamis, M.R., 2021. Does financial support improve the well-being of the 396 elderly?, PROCEEDINGS OF 8TH INTERNATIONAL CONFERENCE ON ADVANCED MATERIALS ENGINEERING & 397 TECHNOLOGY (ICAMET 2020)
Response: We have made changes. Thank you for your careful review.
Reviewer 3 Report
Comments and Suggestions for Authors
Interesting article on a current topic.
In the methods it claims to be a systematic review, however it does not comply with the PRISMA guidelines, namely: it is not identified in the title, it does not evaluate (or justify the non-evaluation) of the methodological quality of the articles.
P-3
In the flow chart, the number of articles must be reviewed: the calculations are not correct.
Figure 1. Flow diagram of the study selection -
181 articles identified considered for full review
Articles abaut MT tests (n=70)
Articles abaut MT training (n= 110)
70+110 ≠ 181
Articles abaut MT tests (n=70)
60 excluded – (70-60≠7)
Articles abaut MT training (n= 110) – 110-98≠11
P. 4
L – 140-143
“A total of 11 studies(Bischoff et al., 2021; Kwan et al., 2021; Mak et al., 2022; Merchant 140 et al., 2021a; Merchant et al., 2021b; Nascimento et al., 2023; Rezola-Pardo et al., 2019; 141 Silsupadol et al., 2009a; Szturm et al., 2011; Trombetti et al., 2011; Zhao et al., 2021) were 142 finally included.”
Articles abaut MT training? should be mentioned in this sentence.
P- 6
“Table 1 Characteristics of the included studies” in testing?
Table 1 and table 2 - Although the acronyms are identified in the subtitles, it is difficult to interpret. The placement of some results should be considered
Table 2 - It must be well detailed which groups existed, which interventions were in each group and especially in the results which group showed the improvements noted.
For example: in the study by Trombetti et al., refers to an experimental group (EG) and a control group (CG). However, it is not explained what intervention in the CG and whether the results are in favor of the GE.
L- 295 AD patients – AD? Must be written in full
Comments on the Quality of English LanguageWriting benefits from shorter sentences and paragraphs.
Author Response
Interesting article on a current topic.
In the methods it claims to be a systematic review, however it does not comply with the PRISMA guidelines, namely: it is not identified in the title, it does not evaluate (or justify the non-evaluation) of the methodological quality of the articles.
Response: We apologize for any unpleasant experience you may have experienced due to our mistranslated wording. This review is a systematic literature search and review of research on dual-task activities in the frail elderly and is not a systematic review(Grant, & Booth, 2009). We have made changes (highlighted in green). In addition, we recognize that the adoption of the PRISMA checklist provides assistance in drafting a high-quality review report, and if you feel this is necessary, we will immediately follow up by adding the appropriate form as an attachment. Thank you again for your careful and patient review!
P-3
In the flow chart, the number of articles must be reviewed: the calculations are not correct.
Figure 1. Flow diagram of the study selection -
181 articles identified considered for full review
Articles abaut MT tests (n=70)
Articles abaut MT training (n= 110)
70+110 ≠ 181
Articles abaut MT tests (n=70)
60 excluded – (70-60≠7)
Articles abaut MT training (n= 110) – 110-98≠11
Response: We are very sorry that the wrong flowchart was placed due to our carelessness. Because several of the authors responsible for the search found problems when analyzing the characteristics of the literature, we repeatedly confirmed the 18 literatures that were finally included, which resulted in the previous flowcharts being inadvertently placed in the formal submission. Thank you for your careful review. We have made changes to the flowchart (highlighted in green).
- 4
L – 140-143
“A total of 11 studies(Bischoff et al., 2021; Kwan et al., 2021; Mak et al., 2022; Merchant 140 et al., 2021a; Merchant et al., 2021b; Nascimento et al., 2023; Rezola-Pardo et al., 2019; 141 Silsupadol et al., 2009a; Szturm et al., 2011; Trombetti et al., 2011; Zhao et al., 2021) were 142 finally included.”
Articles abaut MT training? should be mentioned in this sentence.
Response: We have made changes (highlighted in green). Thank you.
P- 6
“Table 1 Characteristics of the included studies” in testing?
Response: We have made changes (highlighted in green).
Table 1 and table 2 - Although the acronyms are identified in the subtitles, it is difficult to interpret. The placement of some results should be considered
Response: We have adjusted the placement of the contents of Tables 1 and 2 in accordance with your suggestions (highlighted in green). Thank you!
Table 2 - It must be well detailed which groups existed, which interventions were in each group and especially in the results which group showed the improvements noted.
For example: in the study by Trombetti et al., refers to an experimental group (EG) and a control group (CG). However, it is not explained what intervention in the CG and whether the results are in favor of the GE.
Response: Thank you for your valuable advice! We have added and modified Table 2 to make the presentation of the results clearer.
L- 295 AD patients – AD? Must be written in full
Response: We have made changes.
Round 2
Reviewer 1 Report
Comments and Suggestions for Authors
The paper was revised based on comments.
There is no description of the search date, so I don't know when it is, but I think it is out of date, so please update it. If the literature is growing, add it to the text.
Author Response
Thank you for your careful review.We have now updated the search date to November 2023 (2. Materials and methods) (highlighted in blue).
Reviewer 3 Report
Comments and Suggestions for Authors
I appreciate the changes made that help the understanding of the article.
However, to be able to understand table 2 I had to consult the original articles (which should not be necessary - the article should be understood without additional reading).
If there are two groups with two different interventions, it is necessary to clearly indicate which group they refer to in the results.
For example: In the study by Rezola-Pardo et al. [34], 2019 there are two groups with different interventions and the difference between the two groups is not noticeable in the results.
In the study by Kwan et al.[30] 2021, does not indicate which of the groups had the indicated result. The reader is led to think that the experimental group had that result, but to confirm they have to go to the original article.
Comments on the Quality of English Language-
Author Response
Thank you for your positive comments! Based on your suggestions, we have adjusted the presentation of the indicators in Table 2 to give readers a clearer understanding of the characteristics of the included studies(highlighted in blue).